# Using an Interpretive Phenomenological Approach to Understand the Menstrual Experience of Young Adults

**DOI:** 10.3390/nursrep15020065

**Published:** 2025-02-11

**Authors:** Catherine Graeve, Vera Stephenson, Grace Gao

**Affiliations:** School of Nursing, St. Catherine University, Saint Paul, MN 55105, USA; vistephenson@stkate.edu (V.S.); ggao912@stkate.edu (G.G.)

**Keywords:** women’s health, menstrual problems, quality of life, racial disparities, social stigma, dysmenorrhea, culture, time off work/school, menstrual education

## Abstract

**Background/Objectives**: an estimated 1.8 billion people worldwide menstruate, and many face difficulties managing. Young adults often encounter stigma, exclusion, and a lack of resources to manage menstruation comfortably. A review of studies on college students revealed that menstrual symptoms and stigma lead to absenteeism, poorer performance at work and school, and decreased quality of life. This study examines the multifaceted perceptions of a diverse group of young adults aged 18–25 to gain a deeper understanding of the menstrual experience, including cultural stigma, to advocate for personalized care and policy change. **Methods**: we used snowball sampling and employed a qualitative research methodology integrating a Qualtrics survey (n = 620) and focus groups (n = 50). We used an interpretive phenomenological approach to thematically code participants’ lived experiences by assigning codes to text segments and grouping them by broader themes using the Nvivo 14.23.3 software to understand the underlying meaning and significance of the data. **Results**: a diverse group of young adults completed the study. Key themes include difficulties attending work/school/social activities related to the physical and mental health challenges associated with menstruation, a cultural stigma, and a lack of access to healthcare and care products during menstruation. Quality of life could be improved with decreased stigma and improved self-care strategies. Limitations include the small sample size and the qualitative study design. **Conclusions**: this study highlights the need for a holistic approach to supporting menstruation. Recommendations include increasing access to menstrual healthcare, an understanding of cultural differences, and consideration of policy accommodations such as paid time off by workplaces and educational institutions related to menstruation.

## 1. Introduction

Menstruation, while a normal part of life for over 1 billion girls, women, transgender, and non-binary persons globally, is not always straightforward [1]. Issues surrounding menstruation, especially for young adults, include lack of education, stigma, poor access to menstrual hygiene products, missed work and school, and physical and mental health concerns. Menstruation can be associated with pain, heavy menstrual bleeding, and mood-related symptoms [2,3,4]. Many young adults who menstruate report symptoms that significantly change their ability to fully engage in social activities, school, work, and intimate relationships [5,6]. Recently passed “Menstrual Health Day” policies reflect this [5]. Menstruation also continues to carry the stigma associated with it and is often not spoken of [7]. Culture can impact this, but no culture appears immune to the stigma, as cultural behaviors significantly shape human health [8]. Access to period products such as pads and tampons is a concern and creates unnecessary health inequities [9]. While there are quantitative studies of young adults showing an association between increased menstrual symptoms and decreased quality of life [1,3,4,6], there seems to be a gap in the literature surrounding the overall holistic experience of menstruation for young adults. This study asks questions related to a wide array of symptoms of menstruation, quality of life, and how individuals learn about menstruation, followed by focus groups asking more detailed questions about the experience of a diverse sample of young women surrounding menstruation. The objectives for the study include:How do menstrual symptoms impact the quality of life for young adults, including work, school, and social connections?Do menstrual symptoms have a different impact on quality of life for young adults from different racial/ethnic groups?Are there holistic factors that could improve an individual’s ability to cope with these factors, such as clinical and self-care?

The conclusions include eight key themes: menstrual symptoms can be severe and lead to missed activities, individuals often face inadequate medical care for menstrual concerns, menstruation can lower the quality of life, work and school should provide accommodations, self-care measures can alleviate symptoms, cultural differences influence menstrual education, stigma surrounds discussions of menstruation, and there is a need for greater advocacy surrounding “normalizing” the menstrual experience.

## 2. Materials and Methods

While this work was primarily qualitative, using focus groups, we also administered an online survey using Qualtrics. We used interpretative phenomenological analysis (IPA) in the focus groups to explore individuals’ lived experiences and how they interpret these lived experiences within the framework of their personal and social environments [10].

Using flyers, online social media platforms, and snowball sampling, we recruited menstruating young adults aged 18–25 to complete the survey (n = 620). The snowball sampling may have led to an overrepresentation of individuals with more negative menstrual experiences. The survey used multiple choice and descriptive, open-ended questions about symptoms, where respondents learned about menstruation, how their culture impacted their menstrual experience, and quality of life surrounding menstruation. Survey results showed mixed responses to questions regarding education, stigma, and quality of life related to menstrual experiences which informed our focus group questions to explore those areas in more depth. Participants who indicated on the survey that they were interested in a focus group participation were notified by email, and consent was confirmed by completing a letter of consent sent ahead of the focus group. Eight focus groups were held virtually via Zoom (n = 50). They were facilitated by two co-authors and one graduate student research assistant. All identified as female; two were nurse educators with research experience, and the third worked as a public health professional and, at the time, was a recent graduate with a master’s in Public Health. All three have previously facilitated focus groups that led to publications.

There was no previously established relationship between the participants and the facilitators. The participants had a brief introduction via the study consent form to the researchers. Participants were given a gift card for their participation, something which may have motivated them as well as personal interest in the topic. Since we began with snowball sampling, it is possible that those who chose to participate have had a negative experience with menstruation which piqued their interest. As is the case with other focus groups, participant statements may have been influenced by others’ responses to prompts.

Description statistics were used on the multiple-choice questions. Results informed focus group questions. Focus groups were recorded and transcribed. All identifiable data were removed, and participants were given identification numbers. Two authors (CG and VS) coded the responses to questions independently using the Nvivo 14.23.3 software. To identify key themes, an initial open coding process was conducted to identify relevant concepts from the interview transcripts. Thirty initial codes were identified from themes seen in the literature and the authors’ previous work [11] and included codes such as accommodations, advocacy, expense, cravings, culture, and missed activities. These codes were then reviewed and grouped into five broader categories (quality of life, symptoms, barriers, self-care, and accommodations) based on their conceptual similarities, resulting in a set of emergent themes. To ensure consistency, the authors collaborated and discussed discrepancies.

## 3. Results

Overall, our results show racial, ethnic, and gender diversity, as displayed in Table 1. A large number of the study participants were students from Midwest universities, along with young adults from outside the US. While most respondents identified as female, some reported identifying as male (about 5%), and a small number (about 3%) identified as non-binary. The sample, overall, was relatively diverse, with 25% of the respondents being of Hispanic, Latin, or Spanish origin, as compared to 19% of the overall US census according to the US Census Bureau (2020). Just over half of the respondents reported their race as White, almost 13% as Black or African American, about 7% as African, and another 7% as Asian. Less than 5% of the respondents reported in each of the other categories. Our sample for race was also slightly more diverse than the US census data collected in 2020 (58% white; 12% Black, 6% Asian, 1.3% American Indian or Alaskan Native) [12]. 

Table 2 displays the demographics of the focus group participants, who were recruited based on interest and availability as a subset of survey respondents.

Focus group questions, along with the most commonly used codes from the transcripts, are displayed in Table 3.

Table 4 displays descriptive data from the survey when respondents were asked how they learned about menstruation, how comfortable they felt talking about menstruation with family and friends, and how open their culture is in discussing menstruation. Most respondents learned about menstruation from their families (77%). Of those who reported that they did not learn from their families, most reported learning at school (67%), with others learning from the internet and friends (15% and 12%, respectively). Just over half had a positive experience learning about menstruation, about 40% neutral, and just under 7% negative. Slightly over half of the respondents (61%) responded positively about feeling comfortable talking about menstruation with their family, a higher percentage felt comfortable discussing it with friends (75%). Slightly over half of the respondents agreed or strongly agreed that their culture is open to talking about menstruation.

Table 5 includes themes from the open-ended question, “Please comment on how the family you grew up with and/or your culture impacted your menstruation experience.”. Themes were included based on the frequency of similar respondents and relevancy to the topic and included varied responses on support from family, privacy, menstruation as a taboo topic, gender differences in who discusses menstruation, and that culture and religion play a role. Many of these themes were validated in the focus groups.

Eight themes were identified from the focus groups using thematic analysis. Each theme is described below, along with quote(s).

Theme #1:menstruation-related symptoms, including mental health changes, cause significant discomfort.

Symptoms mentioned included severe pain/cramps, heavy bleeding, mood changes, inconsistency in when symptoms would occur, varied lengths of the period, food cravings, nausea, vomiting, and headaches. While most participants discussed pain and bleeding, the discussion around mental health, when it occurred in the focus group, involved more participants speaking up in agreement. One participant mentioned feeling suicidal during the day before her period.

“I don’t make any big decisions. I don’t purchase anything. I don’t make any decisions about my job or my school. I don’t have any big conversations with people, because I am not in a healthy state of mind during that time. I’ve just come to realize during those times I need to just stop. Focus on myself and not do anything I think I might regret later.”“A lot of the things I do lose all of their joy when I’m on my period. It’s that I have to muscle through.”

“It can be kind of debilitating, as I experience different symptoms whether it’s dizziness, nausea, different things that make daily life pretty difficult to manage.”

“With hormones, but also just like how it impacts the way that I interact with others and the emotional side effects of it. The mood swings!”

“I usually get really sick, like I get bad headaches. I feel nauseous.”

Theme #2:menstrual symptoms often result in missed social, school, and work activities.

Most participants mentioned altering their daily activities during their period, and if they were not able to alter due to policy, not being able to do their work to their full capacity. Reasons for this included low energy, not feeling able to socialize due to anxiety, pain from cramps or headaches, having an upset stomach, and feeling concerned about bleeding through menstrual hygiene products.

“I feel down for some time, and also it affects my daily activities and also changes my normal behavior.”“I don’t go to school when I’m having my periods, because I’m afraid that my pad is going to leak because of the heavy flow.”“My biggest challenge is um like I have to skip school or work a lot because of the pain that I have.”

“I also have extreme nausea when I’m on my period. And so it’s difficult to do something without feeling like I’m going to throw up. So I have taken sick days in the past, because things are not going well for me.”

Theme #3:barriers remain for many in accessing adequate care during menstruation.

Barriers included access to menstrual products, access to care for symptoms, lack of health insurance, and menstrual products that do not always work well. Also mentioned was the huge cost of products and their quality and safety. Participants discussed seeking healthcare treatment and being dismissed when asking for solutions.

“I went to my doctor not long ago talking about my anxiety, and, like all my symptoms and all of that and they like ran a couple of tests but it wasn’t even what I asked for and what I was looking for. It was more just like their own personal thoughts based on my family history. So they didn’t even listen to what I was trying to talk about, and I think I know my body pretty well, so that really frustrated me.”“If you go to the doctor they just want you to track your period. There’s a sense of hopelessness that has become a barrier. It’s like, how can I find this information, and will I ever find this information?”“You gotta pay co-pays and stuff, and like some people, can’t afford that like me. So it’s like I would just have to deal with it if anything did come up with my body. and I don’t know how to navigate that.”

Theme #4:the impact of menstruation diminishes the overall quality of life.

Participants mentioned the big impact that menstruation has on their quality of life, including their relationships. Some mentioned the loss of joy they experience during menstruation, and others discussed the resignation that this will not change and nothing will be done. A few participants mentioned feeling unsafe during their period due to how they were treated.

“But in those few days before and during my period I just feel like I want to be done, and I just want everything to stop. So, during those few days I have a very poor quality of life, and because that happens every single month, the time adds up.”“During my menstrual period, my quality of life is very poor because of the heavy bleeding, and these you get cramps, you know it can be so so uncomfortable. Having a heavy flow. You can’t go out. You can’t do so many things.”

“I definitely have a lot lower energy, like when I’m on my period, like about to get it, and so it can make it harder to be motivated to do things like school or work. I can get kind of annoyed more easily at like my friends, and some stuff, and also just like getting cramps also makes your mood worse and like it makes you just like more tired and not wanting to do normal things.”

Theme #5:employers and educational institutions should accommodate menstrual-related absences.

Each focus group discussed struggling to fully participate in their work and/or school during certain days of their period. Many talked about not feeling comfortable asking for a break or a day off and that it would not be understood.

“I barely have the energy to even go to classes or to even go to work. So sometimes I call off, which, like that, definitely impacts me financially.”“I would say it definitely affects me in terms of school, because sometimes I’m not going to class because it’s just so painful that I just can’t concentrate.”“I think a very important one would be accommodations either at school or at work just a more formal I guess statement and opportunity for us to miss work when we’re not feeling well or taking a break. I think that would really help us feel more comfortable within ourselves throughout the day.”

Theme #6:self-care practices can alleviate menstrual symptoms.

Various coping mechanisms were discussed, such as hot baths, showers, teas, pain relievers, yoga, exercise, and ginger. Participants also mentioned listening to what their body needs in terms of cravings as being supportive of their self-care. Other activities discussed as being helpful included talking with friends, relaxing, “stopping” to rest, sleeping more, and being kinder to oneself.

“It’s okay that you’re not going to feel as good. You don’t have to beat yourself up about it, because I know I’m not going to be as productive. And I know that I’m gonna feel really really bad. It’s my try to tell myself it’s okay, that that’s happening. There’s a reason for that. And you don’t have to feel bad about that.”“I like snack a bit more, just because, like my body is like hungrier when I’m on my cycle. and like I try to like stray away from the shameful feeling of like eating a bit more.”

“I use natural remedies like ginger and hot water.”

“Something I like to do is like taking a bath to prevent any sort of discomfort.”

“I take some rest. I may not have an appetite to eat but if I have an appetite, I eat well, I take a rest, then do more exercise, and that helps to relieve the pains I have during my period.”

Theme #7:cultural factors contribute to variations in menstruation experiences.

There was quite a bit of discussion in the focus group about how culture impacts the menstrual experience. Some mentioned being deemed “dirty” or “impure” and not being able to cook or be around men during their menstrual cycle. Others mentioned the role of culture in the inability of women to discuss menstruation with men as part of their culture. Culture also seemed to dictate some self-care measures, such as teas and tinctures, as well as how a person was supposed to deal with their menstrual cycle mentally. Finally, the culture surrounding which type of menstrual product should be used was discussed.

“In black culture a lot of people don’t talk about things like mental health and etc. So, for our menstrual cycles we really don’t talk about that. You deal with it yourself, and if you need help you’ll ask for it. But you usually don’t ask for help because you feel like you’ll be excluded, or people won’t understand you.”“For example in Latin America we don’t use tampons like it’s pretty weird like they do we do have we can buy them but pads are like more the most common thing there and here I don’t have a really big amount of choices in the store for pads because everyone here uses tampons so that’s kind of frustrating in my in my opinion but I would say just like cultural shocking that I have about my about that here coming here.”“You cannot really share things like cooking like kind of as you upset you. You have to offer a kind of sanitation like being alone in a particular area, and there are some things you ought not to do, and the one that you ought to do when you’re so in my culture. When you are menstruating, you have 7 guidelines to follow.”

“Being that I am African American, like my culture, when I went to the High school I learned menstruation is something I shouldn’t discuss with people. I shouldn’t share my menstruation experience with people. So, I had to keep it to myself.”

Theme #8:menstruation remains stigmatized, and further advocacy is needed for equality.

Participants discussed that menstruation should be normalized so that people would not have to hide it. People felt embarrassed hiding tampons or pads on the way to the bathroom. Others discussed that since they do not identify as female but have a period, they are left out of conversations and feel “extra judged”, since people do not recognize that they may be menstruating. Participants discussed how happy they would feel if they felt supported by their families, healthcare providers, and work and school environments.

“I wish getting care for PMS was more socially and medically acceptable because a lot of places you talk to therapist, doctors, they’ll just say, oh, that’s normal. That’s nothing to be concerned about, and they just aren’t really listening to. This is not something I like. It would like to have stopped.”“Because I think part of it is like the education that you receive in schools right like being able to have an open and honest conversation about like what this means, and what it means for you, and being able to like, feel comfortable talking about it like from a young age. So it’s more normalized.”

“The kind of shame that Cis men will put on people with periods. They say things like I could never buy tampons, that’s embarrassing. I am not understanding why you can go by condoms that target. Why would you have any issues with buying somebody else tampons? I’m not understanding the problem here. but I think that also that stigma, like I still sometimes to this day I will hide a tampon in my sleeve if i’m going to the bathroom like in the middle of class, or I work, and I shouldn’t like, be ashamed to carry a tampon to the bathroom like I shouldn’t feel shameful for that. But I think just like I’ve been conditioned to believe that because of bad experiences I’ve had at the hands of this cis-men.”

## 4. Discussion

Theme #1:menstruation-related symptoms, including mental health changes, cause significant discomfort.

The first theme identified in the focus group was the impact of symptoms like pain, bleeding, and anxiety during certain days of the menstrual cycle. These symptoms were severe enough to interfere with their ability to concentrate at school and work. Other studies have backed this research, including one study of about 1800 Spanish women aged 15–49 which found that 73% reported menstrual pain, 40% reported symptoms impacting their daily lives, and 34% would like to have requested sick leave due to menstrual discomfort (although only 17% did ask and most felt there would be consequences to their professional lives if they did) [6]. Participants discussed the impact of their mental health on menstruation, as outlined in the Results Section. Sarwar and Rauf, who looked at individuals’ access to social support, quality of life (QoL), and mental health problems, found that those with menstrual problems experience more mental health issues, suffer more with depressive symptoms, and have a lower quality of life compared to those without such problems (2021) [13]. Other studies suggest that access to medical care, general information, and education can improve mental health status [3,6,9].

Theme #2:menstrual symptoms often result in missed social, school, and work activities.

The second theme that emerged involved participants expressing that they often missed or could not fully engage in school, work, and social activities on certain days of their menstrual cycle. Existing literature also supports this finding, though more research is needed to explore potential solutions. A systematic review of the literature (83 studies) on the impact of menstrual symptoms on those in higher education found that self-reported dysmenorrhea, other physical and emotional menstrual-related symptoms, and menstrual stigma contributed to negative menstrual experiences among students and that these symptoms contributed to absenteeism, impaired participation and concentration, and declined performance [1]. Schoep et al. found that one in three women quit daily tasks due to menstrual symptoms, and 38% of all women reported being unable to perform all their daily activities (2019) [14].

Theme #3:barriers remain for many in accessing adequate care during menstruation.

Participants also shared their experiences of seeking care for their menstrual concerns, often facing dismissal, difficulty making appointments, or limitations due to financial constraints or healthcare coverage. Since women often do not seek treatment for these concerns, many more women would likely try to seek care if they thought they would be heard [14]. A qualitative study to understand the impact of pain during menstruation on adolescents concluded that there is still a need for interventions to help women manage the symptoms [15].

Theme #4:the impact of menstruation diminishes the overall quality of life.

Based on the discussions of the magnitude of menstrual symptoms, it is unsurprising that participants felt an overall lower quality of life. In 2017, a nationwide, cross-sectional, internet-based survey among 42,879 women concluded that the quality of life can be impacted by menstrual symptoms, which are widespread among the general population [14]. Many other studies have concluded similar findings [1,3,4].

Theme #5:employers and educational institutions should accommodate menstrual-related absences.

Participants discussed feeling like they could not work or go to school due to their severe symptoms. This led to discussions on “menstrual day-off” policies. While the United States has no mandatory requirements under state or federal laws for menstrual leave, some countries such as Japan, China, South Korea, Taiwan, and Zambia do [5]. While there are pros and cons to such policies, one recent study used thematic analysis to examine perceptions about the effects of menstrual leave in the United States and found themes such as the policy being supportive of those who menstruate and those who do not menstruate, a desire to know what they would obtain from it, the existence of concerns about the negative impacts of the workplace as well as how this makes women “look” (weak, unable to handle things), and negative attitudes toward those who menstruate, as they should “just deal with it” [16]. Recent studies have outlined workplace policies that support menstrual health and have concluded that there are various benefits and limitations [17,18,19]. A study in Sweden noted that school nurses needed better guidelines for how to treat and support students with menstrual pain [20].

Theme #6:self-care practices can alleviate menstrual symptoms.

Various self-care measures were discussed during the focus groups. One recent systematic review and meta-analysis exploring self-care surrounding menstrual pain noted that, while young adults, in particular, were utilizing self-care strategies, they were not necessarily using the most effective options for pain management [21]. In the United States, many are unprepared for menstruation, lack the resources to comfortably and confidently manage menstrual health, and are not adequately educated to provide self-care for normal pain and other menstrual issues [22]. Studies have shown that various self-care measurements, such as exercise [23], yoga [24], and herbal teas such as chamomile [25], can improve premenstrual and/or menstrual symptoms. Interestingly, a cross-sectional study explored the impact of living alone on menstrual symptoms and found that the intensity of menstruation-related symptoms was greater for students living alone [26]. These included the intensity of craving for food before menstruation, the intensity of the negative effects before and during menstruation, and the intensity of impaired concentration during menstruation [26]. Something not mentioned in the focus group is the impact of social relationships on symptoms, meaning that not being alone could be a self-care or protective measure. The chosen self-care measures varied across individuals and seemed to have a cultural component.

Theme #7:cultural factors contribute to variations in menstruation experiences.

Focus group participants came from diverse cultural backgrounds and highlighted how culture influenced their menstrual experiences. Although there is limited research on this topic, likely due to the taboo surrounding it, it is reasonable to assume that cultural practices would shape menstrual practices, much like they do in other areas of healthcare [27]. Throughout the focus groups, participants mentioned the various ways in which culture played a role in how they learned about menstruation, discussed (or hid) it with others, and how they self-treated their menstrual symptoms.

Theme #8:menstruation remains stigmatized, and further advocacy is needed for equality.

The focus group results and mainstream and peer-reviewed literature highlight menstrual stigma. There is also a stigma surrounding discussions of menstruation and related symptoms, with many menstrual issues going underreported due to cultural factors. Menstrual symptoms affect the daily activities of 38% of women [13]. A study of 492 girls in Nigeria explored the sociocultural dynamics of menstrual education and found that 41% of girls believed that menstruation should not be discussed openly, and about 37% could not afford disposable menstrual products [28]. Healthily managing menstruation can be challenging for individuals, and many adolescents worldwide experience stigma and social exclusion during their periods [29]. Access to sanitary products is one reason why adolescent girls in Ethiopia and other places miss school during their monthly cycles, and data from multiple studies found that girls having more access to “pocket money” positively improved their menstrual hygiene [7]. In a 2023 study conducted in Japan, approximately 25% of the 198 participants reported having difficulty obtaining sanitary products in the past year for economic and non-economic reasons [30]. A 2022 study of Irish college students exploring menstrual pain management found that censorship of menstruation and dysmenorrhea in the media led to difficulties in accessing information and education. Two students mentioned that accessing information about menstruation was even more limited in their home countries of Malaysia and India [31].

In this study, participants also mentioned stigma for those who identify as male and menstruate. Focus group participants discussed not feeling heard because they do not “look like” they would have a period, being judged, and having their symptoms dismissed.

The themes that emerged from the qualitative analysis of the open-ended survey question enriched and complemented the interpretative element of the interpretative phenomenological analysis and allowed for a more nuanced and comprehensive understanding of the phenomenon under study. The six themes generated from the open-ended question of survey data intersected with the eight themes identified from the focus groups. For example, the “Support from Family” survey theme is closely connected to the focus group theme of “Barriers to Properly Caring for Themselves During Menstruation.” Family support can significantly impact a person’s ability to manage menstruation effectively, especially in cultures where discussing menstruation is taboo. Lack of family support might exacerbate barriers to accessing care, proper hygiene, or emotional support. Together, these themes provide a holistic view of the challenges, cultural dynamics, and potential areas for policy and educational intervention related to menstruation. Combining these perspectives can help address menstrual health more comprehensively by considering personal, cultural, social, and systemic factors. We hope this study highlights the importance of equality and empowerment for all individuals who menstruate, regardless of gender identity. Menstrual health is an issue of public health and human rights [30]. The absence of cultural stigma, along with the need for cultural sensitivity and accessible care, should be considered fundamental human rights. Cultural sensitivity and access to care should be basic human rights.

### 4.1. Limitations

As is often the case in research, we studied a sample of individuals who responded to our call for participants who may have been interested due to their extraordinary menstrual health circumstances or already experiencing specific health issues related to their menstrual cycle. This may have led to an overrepresentation of individuals actively seeking solutions or being more aware of their menstrual health. Using only qualitative methods may have created limitations in generalizing the study’s findings to broader populations. We chose to use focus groups instead of one-on-one interviews, which may have limited the ability to explore each participant’s experience in depth. It is also a possibility that there was self-selection bias, and participants who were more inclined to participate in the focus groups may be more comfortable discussing their menstrual cycles and more willing to share detailed information about their menstrual health, resulting in a sample that is not representative of the broader population. Additionally, we did not engage with focus group participants to review and comment on the findings. Instead, we discussed the findings exclusively among the research team members. While we did not exclude young adults from any geographical areas, we advertised the survey more heavily in the Midwest area of the United States, potentially failing to capture the diversity of experiences in different countries or communities. As a result, our sample may not fully represent all groups. However, the diversity of participants in our study helped strengthen our findings.

### 4.2. Implications

Our findings enhance the understanding of menstruation and menstrual experiences by considering the diverse cultural backgrounds and gender identities of individuals. Gaining a deeper understanding of women’s experiences and perceptions of menstruation can inform the development of culturally responsive educational interventions and policies that promote women’s well-being and menstrual health for all. We need to foster a more inclusive, supportive, and healthier society for people who have challenges posed by menstruation by providing education about menstruation starting in elementary schools, reaching students of all genders and making it available in multiple languages, and creating content sensitive to cultural beliefs and practices. Vending machines with free or affordable menstrual products should be placed in high-traffic areas like schools, universities, offices, and public transport stations. Apps, websites, or social media should be used to share information to reach a broader audience, including those with limited access to in-person education, different learning styles, or literacy levels. Colleges could provide support groups, as participants anecdotally told researchers they felt better after talking to others about their menstrual experience. A policy could be implemented requiring healthcare providers to ask individuals who menstruate about their menstrual health during physical exams, starting in adolescence. Menstrual leave should not require documentation or explanation and should be advocated for at schools without proving their need. Workplace policies without fear of discrimination should be explored, using countries that have passed this legislation as examples. Employers who provide time off, educators who recognize the need for extra time, and healthcare providers who advocate for their patients can help normalize the menstruation experience.

### 4.3. Future Research

Future research that embraces diversity and culture is crucial to understand and address menstrual stigma, improving access to menstrual resources and health education. Studies must look at the impact of menstrual stigma on mental health and the toll on anxiety, depression, and low self-esteem. Researchers must focus on barriers to accessing menstrual products in low-income and rural communities and advocate for targeted subsidies for easier access to free and reduced menstrual products. Exploring how different cultural, religious, and societal norms inform menstruation would provide evidence for developing culturally appropriate education and resources. Studies should also focus on the menstrual health challenges of transgender and non-binary individuals to inform and educate current healthcare systems to better serve this population. The overall goal of all research should be to de-stigmatize discussions on menstruation and period products generally and “normalize” the experience, which could provide support and lead to better relationships and an improved quality of life. This important study calls for a deeper understanding of the menstrual experience, emphasizing the need for empathy, policy change, and personalized care to enhance the quality of life for all individuals navigating menstruation.

## Figures and Tables

**Table 1 nursrep-15-00065-t001:** Demographics of survey participants (n = 620).

Characteristic	n	%
**Age Group**		
18–19	119	19.1
20–21	218	35.2
22–23	157	25.3
24–25	126	20.3
**Gender**		
Female	561	91.8
Male	29	4.7
Nonbinary	15	2.7
Prefer to self-describe	6	0.1
**Ethnicity**		
Hispanic, Latin, or Spanish origin	156	25.3
Not of Hispanic, Latin, or Spanish origin	433	70.3
Prefer not to answer	27	4.4
**Race**		
African	46	6.5
American Indian or Alaska Native	28	3.9
Arab American	33	4.6
Middle Eastern	21	3
Hmong	18	2.5
Native Hawaiian or other Pacific Islander	14	2
White	384	53.4
Asian	50	7
Black or African American	90	12.6
Prefer not to respond	22	3.1
Other	6	0.8
**Approximate Yearly Personal Income**		
Under 20,000	326	53.3
20,000–34,999	101	16.5
34,999–49,999	78	12.8
50,000–74,999	63	10.3
75,000–99,999	33	5.4
75,000–99,999	11	1.8

**Table 2 nursrep-15-00065-t002:** Demographics of focus group participants (n = 50; no data on eight participants).

Demographic	n	%
**Gender**		
Female	37	88
Male	2	4.8
Nonbinary	1	2.4
Transgender male	1	2.4
**Ethnicity**		
Hispanic, Latin, or Spanish origin	6	14.3
Not of Hispanic, Latin, or Spanish origin	35	83.3
Prefer not to respond	1	2.4
**Race**		
American Indian or Alaska Native	1	2.4
Arab American	1	2.4
White	26	61.9
Asian	1	2.4
Black or African American	8	19
Prefer not to respond	3	7.1

**Table 3 nursrep-15-00065-t003:** Focus group questions and codes.

Question	Codes/Most Frequent Responses
What is your biggest concern in relation to your menstrual cycle symptoms?	Bleeding, headaches, irregularity, mood, nausea, pain
How do these symptoms influence your relationships, school, and/or work?	Missed activities, missed school, personal relationship impact, quality of life impact—basic needs, joy, resignation
What barriers do you have in receiving treatment or taking care of your menstrual symptoms?	Expense, product access, healthcare system
How does culture and/or your upbringing influence your menstrual experience?	Stigma, influences, hidden, dirty
Discuss healthy self-behaviors/coping mechanisms that you have engaged in or currently engage in to promote healing surrounding menstruation.	Showers/baths/cleaning, physical activity, natural remedies, cravings, medication
How do you perceive quality of life? Do you feel that menstrual symptoms impact your quality of life?	Focus on what is needed, basic needs met Lose joy, missed activities, pain, bleeding, poor quality of life
If there was one thing you could change about your menstrual cycle, what would it be?	Days off during period, not hidden, more support

**Table 4 nursrep-15-00065-t004:** Survey questions on menstruation.

Question	Response	n	%
Did you learn about menstruation from the family you grew up with?	Yes	449	77.15
	No	91	15.64
	Cannot remember	42	7.22
If no, who did you learn about menstruation from?	School	44	66.7
	Internet	10	15
	Friends	8	12
	Other adult	4	6
If yes, was the experience…	Positive	231	52.5
	Negative	29	6.6
	Neutral	180	40.9
I feel comfortable talking about menstruation with the family I grew up with.	Strongly disagree	38	6.6
	Somewhat disagree	96	16.7
	Neither agree nor disagree	82	14.3
	Somewhat agree	195	34
	Strongly agree	163	28.4
I feel comfortable talking about menstruation with my friends.	Strongly disagree	30	5.2
	Somewhat disagree	55	9.6
	Neither agree nor disagree	57	9.9
	Somewhat agree	156	27.2
	Strongly agree	276	48.1
My culture is open to talking about menstruation.	Strongly disagree	48	8.4
	Somewhat disagree	99	17.3
	Neither agree nor disagree	106	18.5
	Somewhat agree	193	33.7
	Strongly agree	127	22.2

**Table 5 nursrep-15-00065-t005:** Summary of open-ended question: “Please comment on how the family you grew up with and/or your culture impacted your menstruation experience.”.

Theme	Example Quote(s)
Family Support	“I had two older sisters that I grew up with so it was something easily talked about” “My parents threw me my first period party and encouraged me to see it in a positive light, even if it made my body feel temporarily weak.” “The family I grew up in was supportive of my menstruation experience which was helpful when I started to experience difficulties with it.”
Privacy	“My mom was very private about menstrual cycles, and so didn’t really teach me anything.” “I didn’t really talk about menstruation much with my mom. I never told her when I got my first period. I never talk about menstruation with my dad or brother.” “I constantly had to hide my pads (used or not) and not talk about it. If I did, it was only with my mom.” “I was terrified when I first got my period and no one told me how to deal with it.”
Taboo Topic	“Menstruation topic was seen as a taboo and was never mentioned in conversations even with parents” “Menstruation was a taboo. I was meant to feel “impure” and gross.” “I believe the societal norms around menstrual cycle are sexist and shameful. For example, when you are in public spaces if you have to go to the bathroom with a tampon you need to hide it or how women don’t openly discuss their menstrual cycle because they are embarrassed or afraid to do so.”
Gender Differences	“I feel like I have the typical experience where the mother is supportive and helpful and the males in the family just don’t want to hear anything about it.” “Can be shy of the males in the family.” “I am very open talking about menstruation with my mother and sisters, but never talk about it in front of my father or brothers.”
Culture	“In my culture we don’t really discuss our menstrual cycle.” “In my culture, when an individual is menstruating, they are seen as “dirty” or “impure”. It is more extreme in the homeland, and my mother doesn’t prescribe to this in her daily life. However, when my grandparents are present or there is a religious event, those who are menstruating are not allowed to participate in certain activities like cooking and touching food, being in prayer and rituals, and eating alongside others.” “In Asian culture, it is important to have warm things like tea or staying warm.” “There was a lot shame around menstruation in the Somali culture, however, my family tried to make it an open and more neutral experience.”
Religion	“My family did not talk about menstruation due to religious beliefs. If it was talked about, it was extremely sexual in nature and was seen as something to hide”. “I think Jews talk about these things more openly—my dad wouldn’t cringe at anything, even though he’s not the most emotional or open person about life in general. He would even cook me chicken liver and beef for the iron losses, so it seemed like he was more educated than most white men. My mom was always super supportive and looked at it kinda spiritually sometimes too” “I grew up in a Christian household so I think they relied on my school to educate me.”

## Data Availability

The survey data are available through Mendeley at https://data.mendeley.com/datasets/c8nwbfrz5d/1 (accessed on 13 December 2024). The focus group data are available upon request.

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
