# Peer review of "Using an Interpretive Phenomenological Approach to Understand the Menstrual Experience of Young Adults"

_nursrep, 2025, doi:10.3390/nursrep15020065_

Round 1

Reviewer 1 Report

Comments and Suggestions for Authors

The abstract includes the aim of the study, which is to understand the menstruation experiences of young adults, and the methods used are stated in general. It is stated that questionnaire and focus group methods were used in the research and the data were analysed with a qualitative approach ‘Interpretive Phenomenological Approach’. The findings emphasised themes such as physical and mental health problems, social stigma, cultural differences and low quality of life. In addition, the importance of personalised approaches in menstruation management and increasing access to health services were mentioned.

However, the summary does not cover all aspects of the study. Details of the methodology are missing; for example, how the data was collected, the software used in the analysis and the details of the process are not included. The findings do not include all the main themes from the focus groups. In particular, the roles of employers and educational institutions, self-care strategies and the influence of cultural factors could have been presented more comprehensively. Limitations of the study, sample characteristics and recommendations for the future are also not included in the summary. Furthermore, it should have been emphasised more clearly how the results of the study can contribute to policy and practice.

The introduction begins by emphasising that menstruation is a global health problem and the importance of the issue is outlined in general terms. Addressing key issues such as lack of education, social stigma, inadequate hygiene products and access to health services provides a comprehensive overview. However, the outdated nature of most of the sources used (e.g. 2011, 2018 and 2019) weakened the timeliness of the study. The literature analysis is not in-depth and the research gap between previous studies and this study is not clearly explained. The unique contribution and difference of the study is not clearly articulated. Furthermore, the focal point, namely young adults' experiences of menstruation, was not directly addressed. In addition, it is an important deficiency that the main research questions of the study are not included in the introduction. Clearly stating the research questions would have made the scope of the study and its objectives clearer for the reader. These deficiencies cause the introduction to fail to convey the importance and scope of the study to the reader in a strong way. The introduction needs to have a more systematic structure, clearly state the research questions and relate it to more recent literature.

The method section explains the general research methods of the study and specifies the data collection tools used. It is stated that the study was conducted with a qualitative method and data were collected through questionnaires and focus group interviews. It is also stated that ‘Interpretive Phenomenological Approach’ was used to analyse the data. However, there are some important deficiencies in the method section. The details of the data collection process are not sufficiently explained. For example, it is not explained how the questionnaire and focus group interviews were structured, according to which criteria the questions were prepared or whether the reliability and validity analyses of the data collection tools were performed. Furthermore, the methods used in conducting the focus group interviews, the selection process of the participants and the detailed protocol of the interviews were not clearly stated. It is not discussed how the sample size and diversity of the study may affect the research findings. The impact of the participants' different backgrounds regarding their menstruation experiences on the analysis was not evaluated.The information provided on the data analysis process is insufficient. The process followed in determining the themes and how this process ensured reliability between independent coders were not explained. In addition, it is not stated whether feedback was received from the participants to improve the accuracy of the themes obtained. Although the ethical aspects of the study, such as the ethical approval process and the consents obtained from the participants, are briefly mentioned, these issues need to be elaborated. 

The findings section clearly set out the main themes of the study and addressed key issues such as physical and mental health problems, social stigma, cultural differences, inadequate access to health services, low quality of life and self-care strategies. The chapter is successful in presenting the general framework of the themes and the fact that it is supported by participant views is a positive aspect. However, some shortcomings are noteworthy. The depth of the themes was not sufficiently addressed and a more detailed analysis of the identified topics was not made. Participant views were presented in limited number, and a stronger representation of the themes could have been ensured by including the views of more participants. Furthermore, the links between the statistical data obtained in the questionnaire and the focus group themes could have been explained more clearly. Despite these shortcomings, the chapter overall provides a structure that supports the purpose of the study. A more comprehensive and in-depth treatment of the findings could enhance the overall impact of the study.

The discussion section was successful in linking the findings of the study with the existing literature. The effects of menstruation on individuals' quality of life were discussed and various recommendations were made in terms of health services and cultural sensitivity based on the themes obtained. Making connections with the literature has increased the scientific value of the chapter. However, some deficiencies stand out. It is seen that the themes identified are not analysed in depth and the effects of each theme on individual and social level are not discussed in more detail. Although the contribution of the study to the literature is stated in general, the unique aspects of this contribution are not sufficiently emphasised.

Limitations are mentioned in the article, but this part is not discussed in sufficient detail. It is stated that the fact that the participants consisted only of people interested in the research limits the generalisability of the results. In addition, it is stated that participants may influence each other in focus group interviews. Apart from this, it is not clear to what extent the methods and questions used reflect the scope of the themes. The use of only qualitative methods may have created limitations in generalising the findings of the study to wider populations.

Finally, the sources are old. New literature should be used more.

Author Response

Thank you for your thorough review and response to our manuscript. We appreciate the suggestions and have incorporated them into our revised manuscript. We have attached the comments/responses below. 

Reviewer 2 Report

Comments and Suggestions for Authors

Thank you for the opportunity to review the manuscript titled "Using an Interpretive Phenomenological Approach to Understand the Menstrual Experience of Young Adults." The study addresses an important and timely topic, providing insights into the lived experiences of menstruating young adults and highlighting key themes such as stigma, cultural influences, and quality of life. Below is a detailed review, organized by sections, with specific comments and suggestions for improvement.

Abstract

The abstract effectively summarizes the study, including its background, objectives, methodology, and results. However, it could benefit from a more concise presentation of key findings and their implications. For example, the abstract could more clearly highlight how the identified themes (e.g., stigma, cultural influences) connect to actionable recommendations for policy and practice.

Introduction

The introduction provides a strong rationale for the study and effectively situates the research within the context of existing literature. However, it would be beneficial to explicitly identify gaps in the literature that this study aims to address. Additionally, a clearer articulation of the study's objectives and research questions would enhance the focus.

Methods

The methodology is well-detailed, and the use of an Interpretive Phenomenological Approach (IPA) is appropriate for exploring the lived experiences of participants. However, there are some areas that require clarification or improvement:

  • The recruitment strategy, particularly the use of snowball sampling, should be discussed in terms of potential biases and limitations. For example, participants with more negative menstrual experiences may have been overrepresented.
  • While the survey and focus groups are complementary, it would be helpful to provide more detail on how the survey results informed the focus group questions.
  • The use of NVivo software for thematic analysis is a strength, but the process for coding and theme development should be described in more detail to enhance transparency and reproducibility.

Results

The results section is comprehensive and effectively presents the key themes that emerged from the data. However:

  • The demographics table (Table 1) is clear, but a discussion of how the sample's diversity (or lack thereof) may influence the generalizability of the findings would be valuable.
  • The themes are well-described, but the inclusion of more direct quotes from participants could enhance the richness of the findings.

Discussion

The discussion provides a thoughtful interpretation of the findings and situates them within the broader literature. Suggestions for improvement include:

  • Expanding on the implications of the findings for practice, particularly in terms of educational interventions and policy changes.
  • Addressing the limitations of the study in greater depth, including the potential impact of self-selection bias and the limited geographic scope.

Conclusion

The conclusion effectively summarizes the study's contributions but could be strengthened by providing clearer recommendations for future research and practice. For example, specific suggestions for addressing menstrual stigma and improving access to resources could be included.

Author Response

Thanks for the excellent comments, which we feel have strengthened our manuscript significantly. Please see the attachment for detailed comments/responses.

Round 2

Reviewer 2 Report

Comments and Suggestions for Authors

Dear authors, thank you very much for allowing me to review the manuscript again. The authors have addressed all the issues raised in the previous review.